# Toxicity of *Bacillus thuringiensis* Strains Derived from the Novel Crystal Protein Cry31Aa with High Nematicidal Activity against Rice Parasitic Nematode *Aphelenchoides besseyi*

**DOI:** 10.3390/ijms23158189

**Published:** 2022-07-25

**Authors:** Zhao Liang, Qurban Ali, Yujie Wang, Guangyuan Mu, Xuefei Kan, Yajun Ren, Hakim Manghwar, Qin Gu, Huijun Wu, Xuewen Gao

**Affiliations:** 1The Sanya Institute of Nanjing Agricultural University, Sanya 572024, China; zhaol0619@163.com (Z.L.); 2020202068@stu.njau.edu.cn (Q.A.); yjiewang57@163.com (Y.W.); renyajun0719@163.com (Y.R.); guqin@njau.edu.cn (Q.G.); hjwu@njau.edu.cn (H.W.); 2Key Laboratory of Integrated Management of Crop Diseases and Pests, Ministry of Education, Department of Plant Pathology, College of Plant Protection, Nanjing Agricultural University, Nanjing 210095, China; 3Shenzhen Batian Ecotypic Engineering Co., Ltd., Shenzhen 518057, China; muguangyuan1@sina.com (G.M.); xavier2014@163.com (X.K.); 4Lushan Botanical Garden, Chinese Academy of Sciences, Jiujiang 332000, China; hakim@lsbg.cn

**Keywords:** *Bacillus thuringiensis*, Cry toxin, nematicidal activity, pore-formation

## Abstract

The plant parasitic nematode, *Aphelenchoides besseyi*, is a serious pest causing severe damage to various crop plants and vegetables. The *Bacillus thuringiensis* (Bt) strains, GBAC46 and NMTD81, and the biological strain, FZB42, showed higher nematicidal activity against *A. besseyi*, by up to 88.80, 82.65, and 75.87%, respectively, in a 96-well plate experiment. We screened the whole genomes of the selected strains by protein-nucleic acid alignment. It was found that the Bt strain GBAC46 showed three novel crystal proteins, namely, Cry31Aa, Cry73Aa, and Cry40ORF, which likely provide for the safe control of nematodes. The Cry31Aa protein was composed of 802 amino acids with a molecular weight of 90.257 kDa and contained a conserved delta-endotoxin insecticidal domain. The Cry31Aa exhibited significant nematicidal activity against *A. besseyi* with a lethal concentration (LC_50_) value of 131.80 μg/mL. Furthermore, the results of in vitro experiments (i.e., rhodamine and propidium iodide (PI) experiments) revealed that the Cry31Aa protein was taken up by *A. besseyi*, which caused damage to the nematode’s intestinal cell membrane, indicating that the Cry31Aa produced a pore-formation toxin. In pot experiments, the selected strains GBAC46, NMTD81, and FZB42 significantly reduced the lesions on leaves by up to 33.56%, 45.66, and 30.34% and also enhanced physiological growth parameters such as root length (65.10, 50.65, and 55.60%), shoot length (68.10, 55.60, and 59.45%), and plant fresh weight (60.71, 56.45, and 55.65%), respectively. The number of nematodes obtained from the plants treated with the selected strains (i.e., GBAC46, NMTD81, and FZB42) and *A. besseyi* was significantly reduced, with 0.56, 0.83., 1.11, and 5.04 seedling mL^−1^ nematodes were achieved, respectively. Moreover, the qRT-PCR analysis showed that the defense-related genes were upregulated, and the activity of hydrogen peroxide (H_2_O_2_) increased while malondialdehyde (MDA) decreased in rice leaves compared to the control. Therefore, it was concluded that the Bt strains GBAC46 and NMTD81 can promote rice growth, induce high expression of rice defense-related genes, and activate systemic resistance in rice. More importantly, the application of the novel Cry31Aa protein has high potential for the efficient and safe prevention and green control of plant parasitic nematodes.

## 1. Introduction

Plant pathogens, such as fungi, bacteria, viruses, and nematodes, cause different plant diseases, resulting in yield loss in crops around the globe [1,2]. Plant parasitic nematodes are currently serious pests, contributing to USD 157 billion in annual agricultural losses worldwide [3]. *Aphelenchoides besseyi* is one of the most damaging plant parasitic nematodes and can cause severe harm to a variety of significant crops and vegetables. The significant rice seed-borne pest, *A. besseyi*, spreads predominantly through infected seeds. This is the cause of rice white tip disease and is widely distributed throughout rice-growing areas [4]. In general, *A. besseyi* causes yield losses ranging from 10% to 30%, and in strongly infested areas, yield losses might reach 50% [5,6].

Rhizobacteria have been widely reported as biological control agents for plant parasitic nematodes. Numerous *Bacillus* species have been studied for their role in reducing nematodes and insect pests [7,8]. *Bacillus subtilis* and *Bacillus thuringiensis* (Bt) are two of the *Bacillus* species that have been investigated against various pathogens, including nematodes [9,10]. Bt strains can produce a number of parasporal crystal proteins (Cry or d-endotoxins), which are poisonous to a variety of insects and pests [11,12]. Bt strains can also produce various crystal protein toxins (Cry5, Cry6, Cry12, Cry13, Cry14, Cry21, and Cry55) during the growth process. These Cry toxins exhibit a substantial biological activity against various insects, nematodes, and other pathogenic pests [13,14].

Currently, the high efficiency and specificity of these toxic proteins enable Bt strains to be safely used in controlling various insect pests of crops and are considered to be green and safe microbial pesticides—making them one of the most successful microbial pesticides [15,16]. These are widely used in global agricultural production, effectively reducing the number of chemical pesticides [17]. In addition, Bt toxin proteins have also been used in the field of plant genetic engineering to control crop pests [18]. For example, transgenic rapeseed containing the Cry1Ac gene can control diamondback moth, hairy bug, and cotton bollworm [19], and transgenic pigeon bean containing the Cry2Aa gene is resistant to pod borers [20]. The Cry3A gene in transgenic spruce showed toxicity to spruce bark beetles, indicating that Bt and its toxin proteins add significant value to biological control [21].

In addition to insects, plant parasitic nematodes also affect the agricultural economy [22,23]. Many studies have demonstrated that Cry proteins produced by Bt strains have nematicidal activity [16,23]. Cry proteins with nematicidal activity mainly include Cry5, Cry6, Cry14, Cry21, and Cry55 [14,24]; among these, the 3D-Cry protein Cry5 and non-3D-Cry protein Cry6 have been reported to kill nematodes [25,26]. In order to find and identify novel Cry toxins with broad-spectrum activity against insects and nematodes, we compared Bt strains GBAC46 and NMTD81 based on genome-wide screening and found three unknown Cry proteins in the GBAC46 strain. Among the identified Cry proteins, Cry31Aa showed the highest nematicidal activity in killing nematodes in the intestinal cell membrane.

Furthermore, the beneficial properties of Bt strains are not limited to their biological control activity; however, a variety of endophytes possess the ability to indirectly or directly promote plant growth and development [27,28]. Bt is one of the most important phosphate-dissolving bacteria [29], which can be insoluble in the soil through enzymatic activity. The phosphate was hydrolyzed into plant-absorbable phosphorus ions [30]. Bt produces siderophores through a nonribosomal synthetic peptide pathway to help plants absorb iron [31,32]. Additionally, it can synthesize plant hormones (such as indole acetic acid (IAA), deaminase, and volatile organic compounds (VOCs) to promote plant growth [33]. Thus, Bt strains can be used as a biofertilizer agent to promote the uptake and transport of plant mineral nutrients.

Therefore, in the current study, we used different *Bacillus thurgensis* (Bt) strains, GBAC46 and NMTD81, which were previously isolated in the Qinghai–Tibetan Plateau, China, and a well-known biological strain, *B. velezensis* FZB42, against the plant parasitic nematode *A. besseyi*. In this study, we identified a novel plasmid-encoded Cry protoxin Cry31Aa produced by Bt GBAC46, which acts synergistically to kill free-living *A. besseyi* nematodes via intestinal damage. We also determined the lethal concentrations at LC_50_ (131.80 μg/mL) for the selected Cry proteins Cry31Aa, Cry73Aa, and Cry40OR and found that the only Cry31Aa had strong nematicidal toxicity against *A. besseyi.* The present work sheds new light on the mechanism of Bt strains and their Cry31Aa proteins that regulate the genes involved in the defense mechanism that *A. besseyi*-infested rice plants use to efficiently control white tip disease in rice. Thus, the toxins produced from the Bt strain could be employed for effective *A. besseyi* control, benefiting both the rice crop and the environment, offering a more sustainable management technique than the overuse of nonspecific chemical nematicides.

## 2. Results

### 2.1. Sequence Analysis of Cry31Aa and Familiar Cry Protein

Bt is a main pathogenic bacterium, the pathogenicity of which is largely dependent on the parasporal crystal protein (Cry) which it produces during biological control activity [13]. In order to find new Cry toxins, we used SeqHunter 2 software to screen the whole genome of the GBAC46 and NMTD81 strains, and the results predicted 3 unknown Cry protein genes in GBAC46, namely, Cry31Aa, Cry73Aa, and Cry40ORF, while in the NMTD81 strain, there were no new Cry proteins observed (Figure 1A). The Cry31Aa, Cry73Aa, and Cry40ORF proteins contained 802, 395, and 568 amino acids and were 90.257, 44.647, and 65.1902 kDa in size, respectively. The Cry31Aa and Cry73Aa, which were 110–338Aa and 24–227Aa, respectively, at the N-terminal, were in the delta-endotoxin insecticidal domain. Furthermore, the Cry73Aa C-terminal amino acids 524–659 were at the carbohydrate-binding module 6 (CBM6) functional domain, which determines the nematicidal specificity by binding with carbohydrates. In addition, the analysis of the amino acid sequences of three Cry proteins in the GBAC46 strain found that the Cry31Aa protein was 65.48%, similar to the Cry70Ba1 protein (Figure 1B). The Cry40ORF protein had more similarity to the Cry48Ab1 protein, but the similarity of the amino acid sequence was 14.26%. The percentage of amino acid sequence between Cry73Aa and MPP46Aa1 protein was the highest, i.e., 11.62%. These three Cry proteins of the GBAC46 strain are novel and unreported Cry proteins. Furthermore, the SWISS-MODEL was used to predict and analyze the three-dimensional structures of the three Cry proteins of the GBAC46 strain. The result showed that the 3D model from the crystal structure of three Cry proteins was rich in α-helix and β-sheet and predict that the respective proteins may be a perforating toxins (Figure 1C).

### 2.2. Expression, Immunoblot, and Purification of the Three Cry Proteins of GBAC46

The RT-PCR analysis was used for the detection of the three Cry proteins from Bt strain GBAC46. The RT-PCR results showed that the three Cry proteins of the GBAC46 strain were normally transcribed when reverse-transcribed cDNA was used as a template and strong bands were detected in the gel. When RNA was employed as a negative template, no bands were obtained, whereas the size was in agreement with the calculated size of the three proteins, Cry31Aa, Cry40ORF, and Cry73Aa, which was 100, 150, and 150 bp nucleotides, respectively (Figure 2A). In addition, the three Cry proteins of the GBAC46 strain were heterologously expressed in BL21 *E. coli* using the pOPT *E. coli* expression vector. The size of the three proteins, Cry31Aa, Cry73Aa, and Cry40ORF, was 104.39, 58.78, and 79.32 kDa, respectively. The results of the Western blot and SDS-PGAE showed that the three Cry protein target bands were correct as shown in Figure 2B.

### 2.3. Virulence of Bacillus Thuringiensis Strains to A. besseyi

The nematicidal activity of the selected Bt strains against *A. besseyi* in vitro was evaluated through a 96-well plate. The plate counting method was used to detect the concentrations of strains being treated with nematodes at different gradients. After 24 h of incubation at 20 °C, the GBAC46, NMTD81, and FZB42 strains showed strong changes in their ability to kill *A. besseyi*, up to 88.80, 82.65, and 75.87%, respectively, compared to the control as shown in Figure 3A. If the nematodes were unable to restore movement after being touched with a needle under a microscope, they were considered dead. In addition, various concentrations in CFU/mL of the selected strains were used. The results showed that with a high concentration in CFU/mL, the mortality rate of *A. besseyi* was 100% compared to the control, ddH_2_O, as shown in Figure 3B.

### 2.4. Morphological Observation and Pore-Formation of Bt Strains Via Staining

*Bacillus* is a typical rod-shaped bacterium. In the late stage of the stable culture period and the dying period, *Bacillus* forms spores in the cell to adapt to the external environment. We stained GBAC46 and NMTD81 strains cultured in SM medium for 48 h with biological staining reagents. The results showed that the vegetative morphology of the GBAC46 and NMTD81 strains were a typical short rod shape, while the spores were nearly elliptical (Figure 4A). Furthermore, a scanning electron microscope (SEM) was used to observe the spore and parasporal crystal protein of GBAC46 and NMTD81 after being cultured in an ICPM medium for 80 h. The outcomes showed that both selected strains formed more spores and Cry as shown in Figure 4B.

### 2.5. Toxicity of Cry Protein to A. besseyi

In this study, we tested the selected Cry proteins, Cry31Aa, Cry40ORF, and Cry73Aa, from GBAC46 to determine the nematicidal activity against *A. besseyi*. The results showed that the Cry31Aa protein had the highest nematicidal activity against *A. besseyi*, and its LC_50_ value was 131.80 μg/mL. In addition, we used rhodamine to label three Cry proteins and detect whether the protein was ingested by the nematodes, and PBS was used as the negative control. The results showed that red fluorescence could be observed in the gut of the nematodes after 4 h of treatment with rhodamine-labeled Cry31Aa protein, whereas no red fluorescence was observed in the negative control treated with PBS (Figure 5A). To verify that the Cry31Aa protein was a perforating toxin, we fed *A. besseyi* with the protein Cry31Aa and stained the nematodes with propidium iodide dye. The results showed that the Cry31Aa protein was treated, and the diffusion degree of propidium iodide was present in the intestine of *A. besseyi* (Figure 5B).

### 2.6. Plant Growth Promotion Traits of Bt Strains in a Pot Experiment

In the present study, the selected strains were used as a treatment, and control water was used as a CK. The rice seeds were soaked in a solution of *A. besseyi* for 36 h, and then seeds were dipped into selected bacterial solutions of OD_600_ = 1.0 and planted for 1 month in a greenhouse under controlled conditions. The result showed that the disease intensity was small, and the “white tip” lesions were white to yellow–brown in the control seedlings. Whereas, after being treated with the bacterial solution of selected strains, the disease symptoms on leaves were significantly reduced (Figure 6A). The length of the leaf lesions was calculated, and the disease index of different treatment groups was found to be significantly reduced up to 33.56% 45.66, and 30.34 by GBAC46, NMTD81, and FZB42, respectively.

Further, the growth of rice seedlings along with other physiological parameters were determined to be enhanced by *Bacillus* strains after treating with *A. besseyi.* It was observed that the selected strains, GBAC46, NMTD81, and FZB42, significantly increased root length by 65.10, 50.65, and 55.60%, shoot length by 68.10, 55.60, and 59.45%, and plant fresh weight by 60.71, 56.45, and 55.65%, respectively, compared to the control water treatment (Figure 6b). In addition, the whole plant of rice was cut into small pieces, the nematodes were isolated from the plants, and the concentration of nematodes isolated from the rice seedlings was calculated. The results showed that the number of nematodes obtained from rice plants treated with the selected strains, GBAC46, NMTD81, and FZB42, and the nematode *A. besseyi* significantly decreased by 0.56, 0.83, 1.11, and 5.04 mL^−1^ seedling^−1^, respectively. In conclusion, these results indicated that the selected Bt strains could significantly overcome the rice white tip nematode disease.

### 2.7. Hydrogen Peroxide (H_2_O_2_) and Malondialdehyde (MDA) Analysis in Rice

To further examine the effects of GBAC46 and NMTD81 strains on the defense response of rice, we detected the contents of hydrogen peroxide (H_2_O_2_) and malondialdehyde (MDA), a product of lipid peroxidation in rice leaves after being treated with the selected strains for 24 h. The results showed that both the GBAC46 and NMTD81 strains could significantly induce the accumulation of H_2_O_2_ and reduced the MDA in rice leaves, revealing that the selected strains could induce the defense response of rice compared to the FZB42-positive and CK-negative controls (Figure 7).

### 2.8. Relative Expression of Defense-Related Genes

The effects of the selected strains on the relative expression of rice plant defense-related (*PR*) genes were studied. The results revealed that the GBAC46 strain significantly enhanced the relative expression levels of all defense-related genes, such as *PBZ1, PR1a, PR4*, and *LOX1*, whereas the *PR1a* genes were highly upregulated compared to other genes. Similar results were obtained after treatment with the NMTD81 strain, and the relative expression levels of *PR*-related genes, *PBZ1*, *PR1a,* and *PR4*, were significantly upregulated, indicating that the selected strains could induce high relative expression of rice defense response-related genes (Figure 8).

## 3. Discussion

Plant parasitic nematodes threaten agricultural production and human health [34]. *Bacillus thuringiensis* is a pathogenic bacterium with a pathogenicity that is largely dependent on the parasporal crystal protein which it produces against nematodes [13]. For the discovery and identification of Bt toxins, whole-genome sequencing has proven to be a helpful and efficient method [35,36]. Cry proteins, Cry21Ha, Cry1Ba, Vip1/Vip2, and -exotoxin, were identified as possible nematicidal factors by genome analysis of Bt 4A4, suggesting that this strain possesses many toxins with potential activity against nematodes method [35]. The discovery of new highly toxic Cry toxins will help in developing new pesticides to effectively control diseases related to different insects and nematodes in agriculture.

In this study, we used SeqHunter 2 software to screen the whole genome of the GBAC46 strain, where three unknown Cry protein genes were found, namely, Cry31Aa, Cry73Aa, and Cry40ORF (Figure 1a). The analysis of the amino acid sequences of the three Cry proteins in the GBAC46 strain revealed that the Cry31Aa protein was 65.48% similar to the Cry70Ba1 protein [37]. Recent studies reported that the Cry70Ba1 and Cry21Ha1 proteins had good insecticidal activity against the Lepidoptera beet armyworm and *C. elegans* [38]. The Cry40ORF protein had more similarity to the Cry48Ab1 protein, but the similarity of the amino acid sequence was 14.26%, within delta-endotoxin Cry1Ac-Domain V domain [39]. Furthermore, the SWISS-MODEL was used to predict and analyze the three-dimensional structures of the three Cry proteins of the GBAC46 strain. The results show that the 3D model from the crystal structure of the three Cry proteins was rich in α-helix and β-sheet and predicted that the respective proteins may be a perforating toxin, in agreement with a previous report [40].

Various Bt strains can effectively control different plant parasitic and animal nematodes [11,41]. Bt has been discovered to have a key function in the control of plant nematodes; it not only prevents eggs from hatching but also has nematicidal activity against juveniles (J2) [42]. Various scientists have reported the role of several Bt species in killing nematodes such as *Meloidogyne incognita, Rotylenchulus reniformis, Heterodera gylcines,* and *Caenorhabditis elegans* [11,36]. In this study, we explored the nematicidal activity of the Bt strains against *A. besseyi* in a 96-well plate in vitro. The selected strains GBAC46, NMTD81, and FZB42 revealed significant differences in their ability to kill *A. besseyi* after 24 h of incubation at 20 °C compared to the control. These findings are consistent with prior research that found *P. pacificus* to be more disease resistant than *C. elegans* [43,44]. Another study also found that the secondary metabolite of the FZB42 strain, plantazolicin, kills *C. elegans* and has insecticidal activity against *D. elegans* [45].

Many studies have shown that Cry proteins are highly toxic to nematodes [46,47]. Bt Cry toxins are almost entirely encoded by plasmids and can be used as nematicidal candidates in Bt DB27 [36]. Interestingly, previous crystals revealed a substantial relationship with spores, a trait known as spore–crystal interaction (SCA) [48]. To explore further, we tested whether Bt GBAC46 produced Cry toxins. After culturing for 80 h in an ICPM medium, the results demonstrated that the GBAC46 strain produced more spores, and the parasporal crystal proteins were detected using SEM. These results reveal strong similarities to the SCA of a rare filamentous Bt strain [49].

Cry5, Cry6, Cry14, Cry21, and Cry55 are among the cry proteins that have reported nematicidal activity [24]. One of the prerequisites for a Cry protein to induce its toxic activity is that the toxin protein can be ingested into the intestine by nematodes. It was found that plant parasitic nematodes can only ingest food through narrow needles, limiting the size of their food [50]. It has also been shown that the ability of nematodes to take up proteins is not only determined by the protein molecular weight but may also be affected by the overall size, shape, and electrostatic charge of the protein [51]. Here, we investigated the Cry31Aa proteins from GBAC46 strains. The Cry31Aa proteins were found to show a strong nematicidal activity against *A. besseyi.* In addition, (6)-rhodamine was employed to label three cells, since carboxytetramethylrhodamine is a biological stain with strong fluorescence when dissolved in DMSO solvent and is commonly used for biological dyes [52]. After 4 h of treatment with the rhodamine-labeled Cry31Aa protein, red fluorescence was visible in the guts of the nematodes, which agrees with the results in [53]. Propidium iodide (PI) is a red fluorescent dye that can enter nematode cells through the holes formed by the Cry protein, staining the nucleus red while cells are intact; it is frequently used to determine whether a Cry protein is a perforating toxin protein [9]. The red fluorescence could be seen in the nuclei of the nematode intestinal cells if the Cry protein was a perforating toxin.

The biocontrol potential of the Bt selected isolates were evaluated for their effective control of the plant parasitic nematode *A. besseyi* under greenhouse conditions. The results showed that the disease severity was small and white tip lesions were white to yellow–brown in the control seedling. Whereas after being treated with a bacterial solution of the selected strains with an OD_600_ = 1.0, the disease symptoms on leaves significantly reduced. A similar result was found against root-knot nematode caused by *M. incognita* [54]. In addition, the growth of rice seedlings and the other physiological parameters were also determined. The results revealed that the root length and plant fresh weight significantly increased compared to the control water treatment, which is in agreement with [53].

Furthermore, *Bacillus* spp. and its products have been shown to stimulate the plant immune response and induce systemic resistance (ISR) and upregulate the pathogenicity-related genes in plants [55,56]. The rapid buildup of H_2_O_2_ is considered to be a significant phytopathogen defense signal in cellular plants [57]. The accumulation of malondialdehyde (MDA) levels are important indicators in plants to combat various environmental stresses [10,58] whereas the decreased level of MDA indicated less membrane damage in plants inoculated with *Bacillus* strains under stress conditions [59]. In the current study, it was also found that the H_2_O_2_ levels were upregulated and the MDA levels were downregulated in a very efficient manner in rice plants treated with selected strains.

Bt can also directly promote plant growth by synthesizing phosphate lysing enzymes [60], siderophores [61], and plant hormones [62,63]. When analyzing the growth-promoting effects of GBAC46 and NMTD81 strains in rice, it was found that the selected strains significantly promoted rice physiological parameters such as root and shoot length [10,28]. Moreover, qRT-PCR experiments showed that the GBAC46 and NMTD81 strains were able to enhance the relative expression of rice plant defense-related (PR) genes. The experimental results showed that the selected strains significantly enhanced the relative expression of the defense related genes in rice plants; our results agree with those in [10,64].

## 4. Materials and Methods

### 4.1. Plasmids, Bacterial Strains, and Cultural Conditions

In the current study, we used *Bacillus thuringiensis* GBAC46 and NMTD81 strains, which were previously isolated at the Laboratory of Biocontrol and Bacterial Molecular Biology, Nanjing Agriculture University, Nanjing, China, along with the biological control strain *Bacillus velezensis* FZB42, which was used as a positive control, and double-distilled water (ddH_2_O), which was used as a negative control (Appendix A Appendix A) [65]. All selected strains were incubated in 1/2 Luria-Bertani (LB) liquid medium (0.5% tryptone, 0.25% yeast extract, and 0.5% NaCl, pH 7.0) or agar plates at 30 °C. The *Escherichia coli (E. coli)* strains DH5α and BL21 (DE3) were cultivated at 37 °C in LB agar plates or broth. The required antibiotic, ampicillin (Amp), was added at a final concertation of 100 µg/mL with shaking for 24 h at 200 rpm, according to [66], with some modifications.

### 4.2. Nematode Inoculum

The original culture of the plant parasitic nematode, *Aphelenchoides besseyi*, was provided by Xuan Wang (Laboratory of Plant Nematology, Nanjing Agricultural University, Jiangsu, China). *A. besseyi* was further cultured on *Botrytis cinerea* grown on a potato dextrose agar (PDA) medium for three weeks at 20 °C [67]. The mixture of various nematode stages was collected with a sufficient amount of ddH_2_O in the Petri dishes. The harvested *A. besseyi* was stored at 5 °C in the refrigerator until it was used in the experiment. In addition, in vivo studies were conducted using eggs obtained from a 25 μm sieve after being rinsed twice with ddH_2_O. The Baermann funnel method was used to gather second-stage juveniles (J2) after incubating the eggs for 5 days at 20 °C [68]. These juveniles (J2) were used to test the effect of Bt strains and their Cry31Aa toxin on *A. besseyi*.

### 4.3. Bioinformatics Analysis

The three *cry* genes were found by protein-nucleic acid alignment based on the whole genome of the Bt strains using SeqHunter 2.0 software which was created by Professor Daolong Dou Lab Nanjing Agricultural University China at 2010 based on Microsoft Visual Basic 6.0. A phylogenetic tree was constructed based on the multiple sequence alignment of the full-length of the three Cry proteins from the GBAC46 strain with known Cry proteins from BPPRC database (available online: https://camtech-bpp.ifas.ufl.edu/, accessed on 25 May 2021) using MEGA X 10.0.2. The alignment of the sequences of the Cry proteins was performed using DNAMAN. The crystal structures of the three Cry proteins were predicted based on protein homology using the SWISS-MODEL (available online: https://swissmodel.expasy.org/, accessed on 25 May 2021).

### 4.4. Expression of Cry31Aa Protein and Purification

The full-length PCR product of the Cry31Aa gene amplified from the GBAC46 strain’s chromosomal DNA using primer Cry31Aa-F and Cry31Aa-R was cloned into the expression plasmid pOPTHis between *BamH I* and *Hind III* sites with N-terminal 6hismaltose-binding protein tag and C-terminal flag tag (for primers see Appendix A Appendix A). Then, the recombinant vector was transformed into BL21 (DE3), and the Cry31Aa protein was purified as described previously in [69], with some modifications. Briefly, the expression plasmid-carrying BL21 (DE3) cells were grown at 37 °C with 200 rpm shaking in LB medium containing 100 µg/mL of Amp until an OD_600_ of 0.5–0.7 was achieved. Isopropyl thio-β-D-galactoside, at a final concentration of 1 mM, was added to flasks holding the cultures. The cells were extracted by centrifugation at 10,000× *g* for 20 min after 16 h of incubation at 16 °C with shaking at 200 rpm.

Using a high-pressure homogenizer, the bacterial pellets were resuspended in buffer A (50 mM Hepes, 300 mM NaCl, 5% glycerol, and 30 mM imidazole buffer at pH 8.0), and the cell debris was removed by centrifugation at 10,000× *g* for 20 min. The recombinant proteins were purified in a single chromatographic stage using a 5 mL HisTrapHP column (GE Healthcare, Milwaukee, WI, USA) at 5 mL/min on an AKTA avant 150 machine (GE Healthcare) [70]. The bacterial cell lysate was loaded onto the column, and non-adherent proteins were removed by washing the column with 20 volumes of wash buffer A. A 30–250 mM imidazole gradient was used to elute the proteins in wash buffer B (50 mM Hepes, 300 mM NaCl, 5% glycerol, and 250 mM imidazole buffer at pH 8.0). The purified enzymes were maintained at −80 °C after salt removal using Millipore Amicon Ultra. The proteins were detected and identified using SDS-PAGE. The protein concentration was determined using the Bradford Protein Assay Kit (P0006, Beyotime Institute of Biotechnology, Shanghai, China). For the Western blot, we followed the manufacturer’s instructions and used method anti-Flag (F1804, Sigma-Aldrich, St. Louis, MO, USA).

### 4.5. Total RNA Extraction and cDNA Synthesis for RT-PCR

Total RNA was isolated from the GBAC46 strain after cultivating in liquid LB medium when the OD _= 600_ reached up to 1.0, according to the Bacterial RNA extraction kit’s procedure (OMEGA Bio-tek, Inc., Norcross, GA, USA). After being treated with DNase-I (Takara Shuzo, Takara, Japan), total RNA was reverse transcribed into cDNA using the HiScript II Q RT SuperMix Kit (Vazyme, Nanjing, China) according to the manufacturer’s instructions and determined using the NanoDrop 1000 Spectrophotometer. In a ABI 7500 Fast Real-Time PCR Detection System, reverse transcriptase RT-PCR was performed using SYBR qPCR Master Mix (Vazyme, Nanjing, China).

### 4.6. Nematode Bioassays

The selected strains GBAC46, NMTD81, and FZB42 were cultured overnight in liquid LB medium, then washed with ddH_2_O 2–3 times, and divided into various concentrations of colony-forming units (CFU/mL), 6.41 × 10^7^, 3.98 × 10^6^, 8.59 × 10^5^, and 7.21 × 10^4^, which were applied against *A. besseyi.* For nematode bioassays, the *A. besseyi* was treated with different concentrations of CFU/mL as mentioned above, and ddH_2_O was applied as a negative control in a 96-well plate at 20 °C. The live and dead J2 were counted to calculate the mortality rate (%) using the microscopic examination.

### 4.7. Determination of Sporulation Formation

The production of spores was studied using sporulation medium (SM). Bt cells (1%) from an overnight culture were transferred to 20 mL SM media and cultured for 40 h at 37 °C with 200 rpm shaking. The spore development was observed in one microliter of the culture. Malachite green and safranine O were employed in the spore staining kit (HB8300-2, Haibo Biotechnology Co., Ltd., Suzhou, China). Malachite green was used to color the spores blue, and safranine O was used to counterstain the living cells red [69]. The experimental setups were carried out in accordance with the manufacturer’s instructions. Using an Olympus BX43 microscope, spores and living cells dyed with various colors were seen using cell Sens Standard Software (Tokyo, Japan).

### 4.8. Phenotypic Observation by Scanning Electron Microscopy

The selected Bt strains were cultivated for 72 h at 30 °C with a 200 rpm shaking time on a liquid LB medium. Centrifugation was used to recover the spore–crystal mixture, and the spore–crystal mixtures (1 mL of liquid culture) were washed three times with 1 M NaCl and ice-cold ddH_2_O and then rinsed 2–3 times with ddH_2_O. The parasporal crystal protein was observed using scanning electron microscopy (SEM). Following the company protocols, the samples were fixed in 2.5% glutaraldehyde and 1% osmic acid, coated with gold particles, and then observed using a Hitachi S-3000N SEM at a voltage of 5 kV (Hitachi, Tokyo, Japan).

### 4.9. Endocytosis Assays with Rhodamine-Labeled Cry31Aa

The endocytosis assays were carried out as previously described in [52], with some modifications. The 5(6)-carboxytetramethylrhodamine is a biological stain with strong fluorescence when dissolved in DMSO solvent and is often used as a biological dye [52]. A of 1 mg/mL of rhodamine was used to label the three Cry proteins to detect whether the protein was ingested by nematodes, and PBS was used as the negative control. After 2 h, the nematode samples were washed three times with sterilized water before being observed with an Olympus fluorescent microscope. An objective lens ×40 was used to capture the images. The rhodamine filter was used to detect fluorescence at the same time.

### 4.10. Propidium Iodide Uptake Assays

Propidium iodide (Sigma) uptake tests were used for pore-formation experiments, according to the methodology reported in [71], with some modifications. Nematodes were observed using a fluorescent microscope after being incubated with Cry31Aa for 4 h and 20 mM propidium iodide for 3 h. The wavelengths of excitation and emission were 555 and 580 nm, respectively. ImageJ was used to process the images.

### 4.11. In Vivo Experiments with Rice Plants

The rice cultivar (Daohuahua no. 4) was grown in a greenhouse in controlled condition. Seeds were surface sterilized by 70% ethanol for 30 s (s), 5% (*w*/*v*) sodium hypochlorite solution for 20 min, and finally washed with ddH_2_O four times. The rice seeds were soaked in ddH_2_O for 36 h to accelerate germination. After 36 h, the rice seeds were dipped into *A. besseyi* nematode solution (nematode concentration of approximately 500 juveniles/mL) and ddH_2_O was used as a control (CK) and placed at room temperature for 36 h. The soil was autoclaved at 140 °C for 40 min. A similar sized pot was filled with sterilized soil and used for sowing rice seeds. The 15 mL of overnight culture of the selected strains (i.e., GBAC466 and NMTD81) and the positive control, FZB42, with an OD_600nm_ = 1.0 was applied in rice seedlings grown in sterilized soil. Then, the rice seeds were cultured for one month in a greenhouse under control conditions.

### 4.12. Estimation of MDA Level (Lipid Peroxidation) in Rice Plants

Lipid peroxidation was examined in terms of the malondialdehyde (MDA) level studied in plant leaves under stress conditions according to [72], with some changes. Briefly, the fresh leaves (0.1 g) of rice plants treated with the selected strains as well as the control plants were taken and homogenized in a 0.1% (*w*/*v*) TCA 500 µL solution. The mixture was then centrifuged at 4 °C and 13,000 rpm. The supernatant of each treatment was then mixed with 1.5 mL of 0.5% TBA solution and heated to 95 °C in water for 25 min. The mixture was placed on ice for 5 min to stop any further reaction. The absorbance of the reaction was analyzed at 532 and 600 nm using a microplate reader (Spectrum max plus; Molecular Devices, Sunnyvale, CA, USA).

### 4.13. Determination of Hydrogen Peroxide

Rice leaves extracts were tested for hydrogen peroxides (H_2_O_2_) concentration. Fresh rice plant leaves (0.1 g) were homogenized at 4uC in a 1:9 (*w*/*v*) phosphate buffer (50 mM, pH 6.0). A hydrogen peroxide assay kit (Beyotime, Shanghai, China) was used to determine the amount of H_2_O_2_ in the leaf sample. The test tubes containing 50 mL test solutions were left at room temperature for 30 min before being measured using a spectrometer at 560 nm. The absorbance measurements were calibrated using a standard curve with known H_2_O_2_ concentrations [72].

### 4.14. Defense-Related Genes’ Expression in Rice Plants

The expression profile of the defense genes (i.e., *PR1a, PR4, LOX1, PBZ1*, and *PAL1*) was carried out through RT q-PCR in rice plants grown after being treated with selected strains in a greenhouse experiment. For this, the selected gene sequences were taken from NCB1, followed by designing primers through the PrimerQuest tool; the primers are listed in the Appendix A Appendix A. The housekeeping gene elongation factor 1-alpha (*ef1*) was used in the present study. Briefly, RNA was extracted from fresh rice plant leaves inoculated with selected strains and ddH_2_O was used as the control grown under infested and non-infested *A. besseyi* in greenhouse conditions after 4 days’ post-inoculation (dpi) through the TRizole method. The Vazyme HiScript II Q RT SuperMix Kit (Vazyme, Nanjing, China) was used for cDNA synthesis. RT-qPCR was performed to analyze the expression profile of selected genes in rice plants through a ABI 7500 Fast Real-Time PCR Detection System (Thermo Fisher Scientific, San Jose, CA, USA). The PCR machine was programmed using the following steps: initial denaturation at 95 °C for 30 s, including 40 cycles of 95 °C for 5 s, and 34 s at 60 °C. Finally, relative quantification was performed according to the comparative C method of 2^−ΔΔCT^ as described in [73].

### 4.15. Statistical Analysis

The data were analyzed using one-way analysis of variance (ANOVA) in SPSS version 26 software and reported as the mean ± standard deviation from three biological replicates. For group comparisons, Fisher’s LSD test was utilized. Outcomes were considered significant at *p* < 0.05.

## 5. Conclusions

In conclusion, the selected Bt strains from the Tibet region, China, showed high nematicidal activity against *A. besseyi*. Three proteins (i.e., Cry31Aa, Cry73Aa, and Cry40ORF) were also characterized and functionally assessed for food safety. The novel Cry protein, Cry31Aa, possessed highly efficient and selective nematicidal activity. Furthermore, the current study adds new insights into the mechanisms by which nematicidal Bt regulates defense-related genes and improves plant growth promotion parameters in rice as well as contributes to the control of plant parasitic nematode diseases.

## Figures and Tables

**Figure 1 ijms-23-08189-f001:**
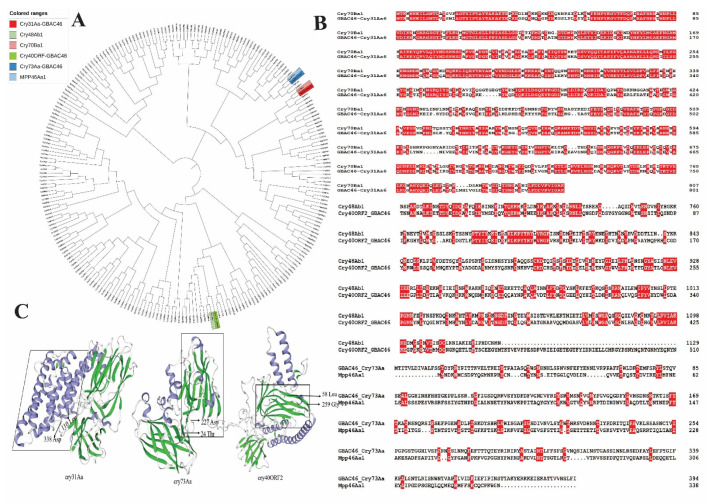
Phylogenetic tree construction was based on the sequence alignment of full-length Cry31Aa and familiar Cry protein (**A**); the amino acid sequences alignment of the Cry31Aa, Cry73Aa, and Cry40ORF proteins (**B**); three-dimensional structure prediction of the three Cry proteins in the GBAC46 strain (**C**).

**Figure 2 ijms-23-08189-f002:**
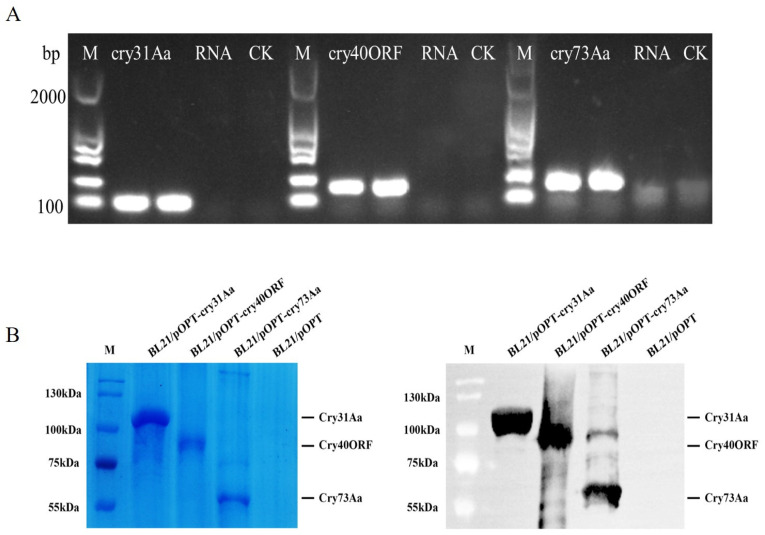
Detection of Cry protein genes and the identification of BL21 *E. coli* using the pOPT *E. coli* expression vector. (**A**) Detection of the transcription of *Cry31Aa* through RT-PCR. Total RNA was extracted from GBAC46 and BL21/pOPT. cDNA was used as a positive control, and RNA was used as a negative control. (**B**) SDS-PAGE expression analysis of Cry proteins (i.e., Cry31Aa, Cry40ORF, and Cry73Aa) after IPTG induction at 16 °C for 16 h. Levels of the three Cry proteins encoded by BL21/pOPT detected via Western blotting. The experiments were repeated independently three times.

**Figure 3 ijms-23-08189-f003:**
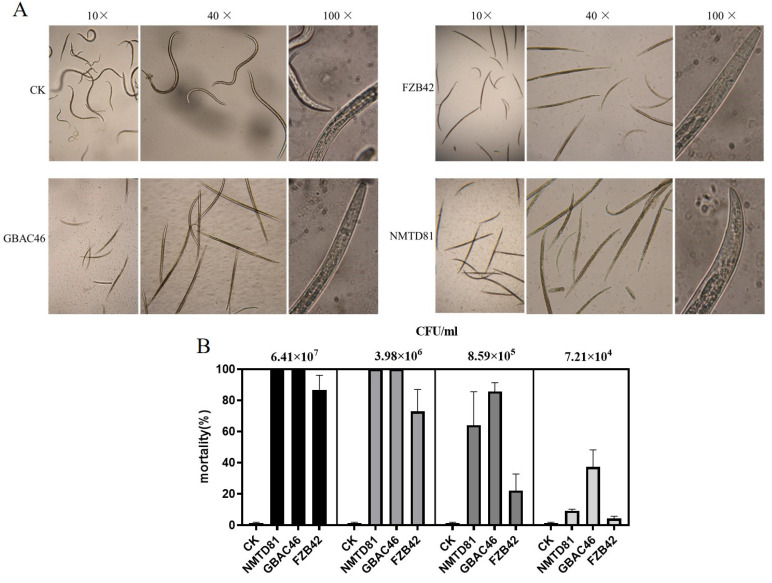
Nematicidal activity of selected *Bacillus* strains against *A. besseyi* compared with the control: (**A**) microscopic figures under different objectives: 10×, 40×, and 100×; (**B**) graphical details of the nematicidal activity of selected *Bacillus* strains. Different error bars show the different mean standard deviation of each treatment repeated three times with triplicates.

**Figure 4 ijms-23-08189-f004:**
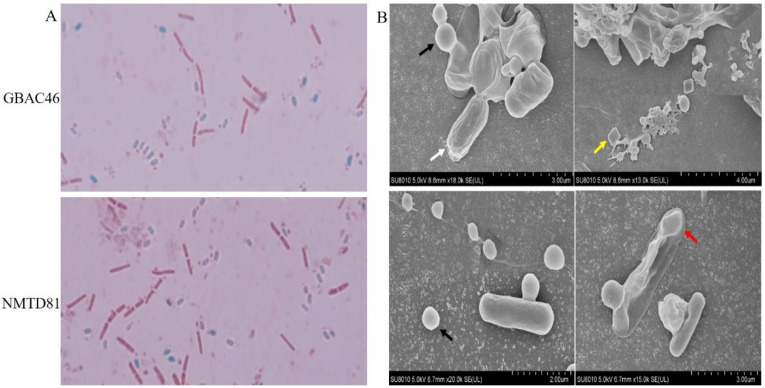
Detection of spore formation in GBAC46 and NMTD81 strains: (**A**) different colors represent different cells: blue—spores, red—living cells under the microscopic study; (**B**) observation of spore and crystal proteins in selected strains under a SEM. White arrow—spores; black arrow—circular crystal proteins; yellow arrow—bipyramidal crystal proteins; red arrow—irregular crystal proteins.

**Figure 5 ijms-23-08189-f005:**
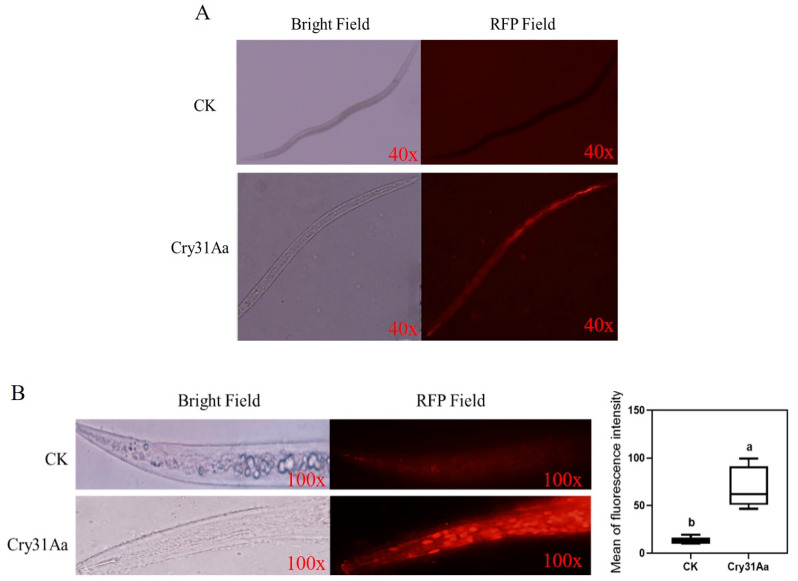
Observation of Cry proteins absorbed by *A. besseyi*: (**A**) the nematode *A. besseyi* after being treated with PBS as a control and rhodamine-labeled Cry31Aa were observed in bright and red fields; (**B**) the Cry31a protein produced the pore-formation toxin exposed to *A. besseyi* after staining with propidium iodide (PI), and fluorescence microscopy was used to monitor the signal of PI. The bright and red fields clearly indicate the changes in the intestinal cells. Different letters indicate significant differences at *p* < 0.005.

**Figure 6 ijms-23-08189-f006:**
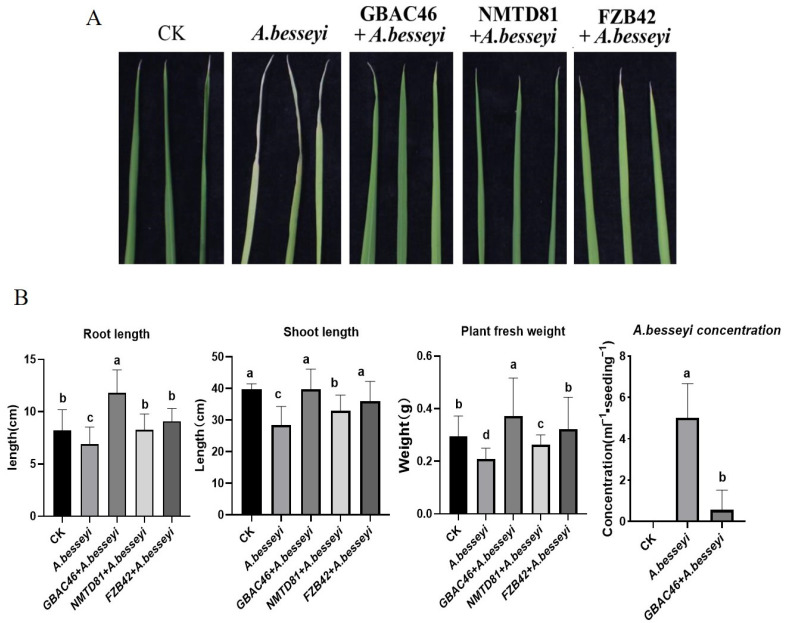
The effect of various *Bacillus* strains against *A. besseyi*-infested rice plants in a greenhouse experiment. The photographic representation of the effect of the selected strains on the infested rice plants (**A**). The rice growth promotion parameters, such as root and shoot length, plant fresh weight, and the detection of several *A. besseyi* nematodes from rice seedlings, after being treated with the selected strains (**B**). The error bars indicate the mean standard deviation of each treatment when replicated three times. Significant differences are indicated by the letters above the columns. Fisher’s LSD test was used to determine significant differences between treatments at *p* ≤ 0.05.

**Figure 7 ijms-23-08189-f007:**
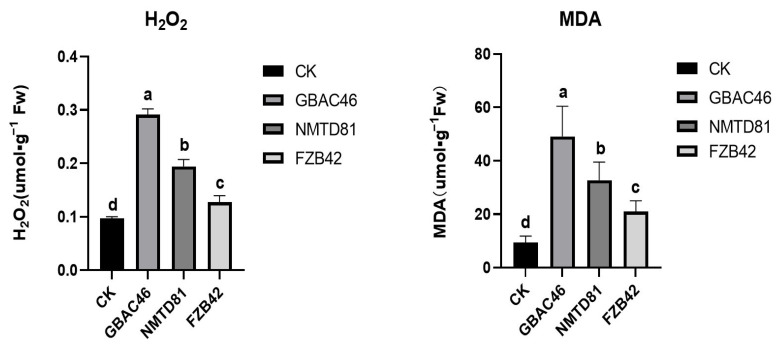
Determination of H_2_O_2_ and MDA in the leaves of 30 day old seedlings following different treatments. The error bars indicate the mean standard deviation of each treatment when replicated three times. Significant differences are indicated by the letters above the columns. Fisher’s LSD test was used to determine significant differences between treatments at *p* ≤ 0.05.

**Figure 8 ijms-23-08189-f008:**
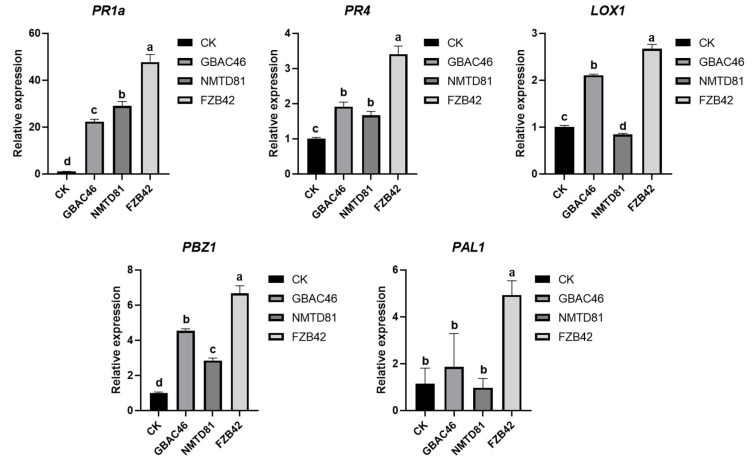
Relative expression analysis of defense genes *PR1a, PR4, LOX1, PBZ1,* and *PAL1* in rice plants infested with *A. besseyi* and treated with Bt strains. The error bars indicate the mean standard deviation of each treatment when replicated three times. Significant differences are indicated by the letters above the columns. Fisher’s LSD test was used to determine significant differences between treatments at *p* ≤ 0.05.

## Data Availability

Not applicable.

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
