# Peer review of "Toxicity of Bacillus thuringiensis Strains Derived from the Novel Crystal Protein Cry31Aa with High Nematicidal Activity against Rice Parasitic Nematode Aphelenchoides besseyi"

_ijms, 2022, doi:10.3390/ijms23158189_

Round 1
Reviewer 1 Report
In the current study, Zhao et., al have briefly demonstrated the Bacillus thuringiensis (Bt) strains GBAC46 and NMTD81 against A. besseyi in the 96-well plate experiment. Furthermore, the authors also screened whole genomes of selected strains by protein-nucleic acid alignment and found that Bt strains GBAC46 showed three novel Cry proteins, namely Cry31Aa, Cry73Aa, and Cry40ORF proteins, which likely provide the safest control of the nematodes. The study topic is very interesting, novel, and helpful for the readerships of IJMS. The overall manuscript is well organized and well-written. However, the main concerns about this manuscript can be found below, and minor revision is suggested.
Lines (15-16), please re-write for better understanding.
Lines (28-29), please check and re-write the sentence.
Line (47-48) incomplete sentence re-write.
Lines (92-94), re-write.
The figure 1 is not very clear; please use a high-resolution figure.
Figure 2 legend is not clear; please recheck it.
In result, lines (196-198) are not clear; please rewrite them for better understanding.
Lines (291-292) not clear, please rewrite.
In line (397), please delete repetition (after after delete one).
Material and methods section 4.5, Authors haven’t cited any references. Similarly, no reference in section 4.6. Add suitable references which you follow protocols.
Lines (449-451) are not clear; please rewrite.
The scientific names should be italic in the whole manuscript; please check carefully.
Abbreviations should be written as a full term (abbreviation) when used for the first time in the text.
Author Response
Dear Editor,
In the current study, Zhao et., al have briefly demonstrated the Bacillus thuringiensis (Bt) strains GBAC46 and NMTD81 against A. besseyi in the 96-well plate experiment. Furthermore, the authors also screened whole genomes of selected strains by protein-nucleic acid alignment and found that Bt strains GBAC46 showed three novel Cry proteins, namely Cry31Aa, Cry73Aa, and Cry40ORF proteins, which likely provide the safest control of the nematodes. The study topic is very interesting, novel, and helpful for the readerships of IJMS. The overall manuscript is well organized and well-written. However, the main concerns about this manuscript can be found below, and minor revision is suggested.
Response; Thank you very much for your helpful suggestions and valuable input in our research manuscript. We also very much appreciate the comments/suggestions made by referees. According to the suggestions, whole manuscript has been revised carefully. We also incorporated most of the suggestions during our revision. A point-by-point response is provided below. The revisions are highlighted in the main text with red color and track changes.
Lines (15-16), please re-write for better understanding.
Response: Thanks a lot, we have modified it.
Lines (28-29), please check and re-write the sentence.
Response: Thanks, it has been modified.
Line (47-48) incomplete sentence re-write.
Response: Thank you so much, it has been modified.
Lines (92-94), re-write.
Response: Thanks for your suggestion, the sentences has been rewritten.
The figure 1 is not very clear; please use a high-resolution figure.
Response: Thank you so much, figures 1 has been changed with high resolutions.
Figure 2 legend is not clear; please recheck it.
Response: Thanks a lot, we have modified the figure legend 2.
In result, lines (196-198) are not clear; please rewrite them for better understanding.
Response: Thanks for your suggestion, the sentences lines has been rewritten.
Lines (291-292) not clear, please rewrite.
Response: Thank you so much, it has been modified.
In line (397), please delete repetition (after after delete one).
Response: Thanks, changes have been made.
Material and methods section 4.5, Authors haven’t cited any references. Similarly, no reference in section 4.6. Add suitable references which you follow protocols.
Response: Thanks, we have modified it according to your suggestion and added new references.
Lines (449-451) are not clear; please rewrite.
Response: Thanks, this sentence has been modified.
The scientific names should be italic in the whole manuscript; please check carefully.
Response: Thank you very much for your valuable suggestions to improve our manuscript. Whole manuscript has been revised accordingly.
Abbreviations should be written as a full term (abbreviation) when used for the first time in the text.
Response: Thanks, abbreviations have been as full term when it was used first time in the text. A full abbreviation list has been provided in supplementary file.

Reviewer 2 Report
The article entitled “Toxicity of Bacillus thuringiensis strains derived novel crystal protein Cry31Aa with high nematicidal activity against parasitic rice nematode Aphelenchoides besseyi” is an original research article. The article is well written and will contribute to the sciences as well. The abstract is well written. The introduction is enough for this study and well-written. The results and discussion is well presented and enough. The discussion may need a comparative study with the previous research on the species and their allies. Some of the discussions section is missing comparative research.Author Response
Dear Editor,
The article entitled “Toxicity of Bacillus thuringiensis strains derived novel crystal protein Cry31Aa with high nematicidal activity against parasitic rice nematode Aphelenchoides besseyi” is an original research article. The article is well written and will contribute to the sciences as well. The abstract is well written. The introduction is enough for this study and well-written. The results and discussion is well presented and enough. The discussion may need a comparative study with the previous research on the species and their allies. Some of the discussions section is missing comparative research.
Response; Thank you very much for your helpful suggestions and valuable input in our research manuscript. We also very much appreciate the comments/suggestions made by referees. According to the suggestions, whole manuscript has been revised carefully. We also incorporated most of the suggestions during our revision. A point-by-point response is provided below. The revisions are highlighted in the main text with red color and track changes. We hope the revised version is more suitable for the publication in IJMS.

Reviewer 3 Report
The authors are evaluating several Bt strains and their respective novel Cry proteins for activity against A. besseyi. This is a very interesting and valuable line of research. however, the manuscript requires considerable editing for English and this reviewer has a number of questions and comments that need to be addressed before publishing.
Lines17-18: Do not just state "better than control" please state how much or LC/EC50 values
Line 20: What do the authors mean by "safest"
Lines 22-23:An LC50 of 131.8 microgram/ml is not that great, but does show some toxicity. Compared to lines 92 and 197, do the authors mean LC50 or EC50?
Lines 25-26: Plese describe the pot experiments in more detail. Were these with rice plants? What species/cultivar was used?
Line 27: Please provide values when stating "reduced infestation"
Line 51: What do the authors mean by "worm species"?
Lines 50-55: What do the authors mean by describing Cry toxins as virulence factors?
Lines 102-103: What was the rationale for even testing GBAC46 for nematicidal activity?
Lines 103: What do the authors mean by "antagonistic activity"?
Line 112: How were the Cry names derived? Was this from the BPPRC?
Line 151: How are the readers supposed to interpret the results of sing CFU/ml?
Lines 193-196: Are the Materials and Methods for this result present in M/M?
Lines 199-202: How were the rice plants treated with GBAC46? What does OD600=1.00 mean? Results state that only 33.56% of the disease is controlled. Please put these results in context
Lines 329-337: It appears that BGAC46 and the Cry proteins have different impacts on overall control of A. besseyi. What is this overall value (perhaps as it impacts yield?)
line 344: Please include a citation for this statement
line 358: Please include a citation for the Baerman funnel method.
For Materials and Methods for 4:13: Please provide more detail
Author Response
Dear Editor,
The authors are evaluating several Bt strains and their respective novel Cry proteins for activity against A. besseyi. This is a very interesting and valuable line of research. however, the manuscript requires considerable editing for English and this reviewer has a number of questions and comments that need to be addressed before publishing.
Response; Thank you very much for your helpful suggestions and valuable input in our research manuscript. We also very much appreciate the comments/suggestions made by referees. According to the suggestions, whole manuscript has been revised carefully. We also incorporated most of the suggestions during our revision. A point-by-point response is provided below. The revisions are highlighted in the main text with red color and track changes.
Lines17-18: Do not just state "better than control" please state how much or LC/EC50 values
Response: Thanks, the sentence has been modified and data are provided.
Line 20: What do the authors mean by "safest"
Response: Thanks, we have changed.
Lines 22-23: An LC50 of 131.8 microgram/ml is not that great, but does show some toxicity. Compared to lines 92 and 197, do the authors mean LC50 or EC50?
Response: Thanks, the sentences has been modified. In addition, in study we used lethal concentration LC50 not EC50.
Lines 25-26: Please describe the pot experiments in more detail. Were these with rice plants? What species/cultivar was used?
Response: Thanks, we have described the pot experiment in details see M/M.
Line 27: Please provide values when stating "reduced infestation"
Response: Thanks, the result has been provided.
Line 51: What do the authors mean by "worm species"?
Response: Thanks, the sentence has been modified.
Lines 50-55: What do the authors mean by describing Cry toxins as virulence factors?
Response: Thanks, we have modified it.
Lines 102-103: What was the rationale for even testing GBAC46 for nematicidal activity?
Response: Thanks, we have changed the sentences and added the nematicidal activity.
Lines 103: What do the authors mean by "antagonistic activity"?
Response: Thanks, the sentence has been modified.
Line 112: How were the Cry names derived? Was this from the BPPRC?
Response: Thanks, Yes, and we already mentioned in material and methods.
Line 151: How are the readers supposed to interpret the results of sing CFU/ml?
Response: Thanks, the sentence has been modified. For detail plate counting method was used to detect the concentrations of strains treated with nematodes at different gradients.
Lines 193-196: Are the Materials and Methods for this result present in M/M?
Response: Thanks, we have provided the pot experiments in detail see M/M.
Lines 199-202: How were the rice plants treated with GBAC46? What does OD600=1.00 mean? Results state that only 33.56% of the disease is controlled. Please put these results in context
Response: Thanks, the results has been described in the percentage form, and data values are provided.
Lines 329-337: It appears that BGAC46 and the Cry proteins have different impacts on overall control of A. besseyi. What is this overall value (perhaps as it impacts yield?).
Response: Thanks, we have changed the sentence.
line 344: Please include a citation for this statement
Response: Thanks, the citation has been added.
line 358: Please include a citation for the Baerman funnel method.
Response: Thanks, we have cited the relative reference.
For Materials and Methods for 4:13: Please provide more detail
Response: Thanks, the section 4:13: has been modified and detail is provided.

Round 2
Reviewer 1 Report
The revised version of the MS is now good but there are many issues in the references.
In main text, references must be numbered as per IJMS guidelines. Please carefully check and correct it in final proof.
Reviewer 3 Report
This is a resubmitted manuscript evaluating the potential for various Bt strains and toxins to control A. besseyi. It appears that the authors have tried to address this reviewer's comments which is appreciated. However, there are just too many English and grammatical errors in this current version for this reviewer even to address in the review (even reading the Abstract was "rough"). So this reviewer recommends that the authors can have someone critically review this MS for scientific English.